# Impacts of Public Debates on Legalizing the Same-Sex Relationships on People’s Daily Lives and Their Related Factors in Taiwan

**DOI:** 10.3390/ijerph17228606

**Published:** 2020-11-19

**Authors:** Huang-Chi Lin, Yi-Lung Chen, Nai-Ying Ko, Yu-Ping Chang, Wei-Hsin Lu, Cheng-Fang Yen

**Affiliations:** 1Department of Psychiatry, School of Medicine, College of Medicine, Kaohsiung Medical University, Kaohsiung 80708, Taiwan; cochigi@yahoo.com.tw; 2Department of Psychiatry, Kaohsiung Medical University Hospital, Kaohsiung 80708, Taiwan; 3Department of Healthcare Administration, Asia University, Taichung 41354, Taiwan; elong@asia.edu.tw; 4Department of Psychology, Asia University, Taichung 41354, Taiwan; 5Department of Nursing, College of Medicine, National Cheng Kung University, Tainan 70101, Taiwan; nyko@mail.ncku.edu.tw; 6School of Nursing, The State University of New York, University at Buffalo, Buffalo, NY 14260, USA; yc73@buffalo.edu; 7Department of Psychiatry, Ditmanson Medical Foundation Chia-Yi Christian Hospital, Chia-Yi City 60002, Taiwan

**Keywords:** homosexuality, minority stress, same-sex relationship, sexual orientation

## Abstract

This study examined the proportion of the individuals who experienced negative impacts on daily lives resulted from public debates on the legalization of same-sex relationships and related factors in Taiwan. Data provided by 1370 participants recruited through a Facebook advertisement were analyzed. Participants completed an online questionnaire assessing negative impact of public debates on daily lives, gender, age, sexual orientation, the number of lesbian, gay and bisexual (LGB) friends, and perceived population’s acceptance of homosexuality. The results showed that 39.5% of participants reported the negative impacts on their occupational or academic performance; 34.2% reported the negative impact on friendship; 37.7% reported the negative impact on family relationship; and 57.4% reported the negative impact on mood or sleep quality. Non-heterosexual participants were more likely to report negative impacts of public debates on all domains of daily lives than heterosexual ones. The number of LGB friends was positively associated with negative impacts of public debates on all domains of daily lives. Participants who were 20–29 years old were more likely to report negative impacts of public debates on the domains of family relationship and mood/sleep quality than those who were 40 or older. Participants who were 30–39 years old were more likely to report negative impacts of public debates on the domain of mood/sleep quality than those who were 40 or older. Males were less likely to report the negative impact on their mood/sleep quality than females. Perceiving population’s acceptance for homosexuality were negatively associated with negative impacts of public debates on the domains of occupational/academic performance, family relationship and mood/sleep quality.

## 1. Introduction

### 1.1. The Ban on Same-Sex Relationships

The ban on same-sex relationships is a structural-level discrimination that denies non-heterosexual individuals, including lesbian, gay, bisexual (LGB), pansexual, asexual, and unsure individuals, legal, financial, health, and other benefits associated with marriage [1,2,3]. The ban on same-sex relationships can negatively affect health of non-heterosexual people. For example, LGB individuals living in the states of the United States that banned gay marriage had increased rates of mood disorder, generalized anxiety disorder, alcohol use disorder, and psychiatric comorbidity, whereas these psychiatric disorders did not increase significantly among LGB respondents living in states without such constitutional amendments [4]. Moreover, the ban on same-sex relationships was one cause of suicidal behaviors in men who have sex with men in China [5]. In total, 28 countries and regions have legalized same-sex relationship in the past three decades [6]. The results of previous studies have indicated that legalizing same-sex relationships has a positive effect on the health of LGB individuals, including reduced discrimination [7,8] and psychological, social, and health benefits to non-heterosexual individuals [9,10].

### 1.2. Advocations of Supporters and Opposers in Public Debates on Legalizing Same-Sex Relationships

As a subject of law and public policy, the legalization of same-sex relationships always evokes intense debates between its supporters and opposers. The most prominent supporters of the legalization of same-sex relationships are human rights organizations and medical and scientific communities. Human rights organizations argue that legalizing same-sex relationships is a civil right; the ban on legalizing same-sex relationships is a violation of human right similar to the ban on interracial marriages [11]. The medical and scientific communities stated homosexuality is a natural and normal human sexuality [12]; gay people form stable and committed relationships that are essentially equivalent to heterosexual relationships [13]. They argue that same-sex parents are as capable as opposite-sex parents to raise children, and the children of same-sex couples fare just as well as those of opposite-sex couples [14]. Moreover, they argue on the public health implications of legalizing same-sex relationships [15,16]. The most prominent opponents are religious groups. Opposition to same-sex relationship is based on claims such as homosexuality is unnatural and abnormal, legalizing same-sex unions will promote homosexuality in the society, and children are better off when raised by opposite-sex couples [17,18]. These debates on legalizing same-sex relationships are prevalent in mass media and social media, especially when bills supporting or opposing same-sex unions are proposed in the legislature.

### 1.3. Public Debates on Legalizing Same-Sex Relationships in Taiwan

Campaigners for sexual minority rights in Taiwan have strived for the legalization of same-sex relationships since the 1980s. Article 972 of Taiwan’s Civil Code poses a problem for same-sex relationships by stipulating that “An agreement to marry shall be reached between a male and a female party of their own accord.” Sexual minority right campaigners appealed for recognizing same-sex relationships, but the Court turned down the petition in 1986 on the grounds that “homosexuality corrupts social values” [19]. In October 2016, a group of legislators proposed a Marriage Equality Bill to change the Civil Code for legally recognizing same-sex relationships and passed its first court reading [20]. Although the Marriage Equality Bill failed to be further considered by the Judiciary and Organic Laws and Statutes Committee because of a lack of support from the ruling party and the main opposition party in Legislative Yuan, it initiated public debates between supporters and opposers of the legalization of same-sex relationships. Public debates not only were carried out in public hearings held by Legislative Yuan but also spread like wildfire in mass and social media. Most of the Taiwanese people paid attention to the topic of legalizing same-sex relationships for the first time [21].

In May 2017, Taiwan’s Council of Grand Justices announced that the current Civil Code that barred same-sex relationships was a violation of human rights to equality and was unconstitutional (Interpretation No. 748). Moreover, it stipulated that same-sex relationships should be legalized in Taiwan within 2 years. In response to the decision of the Council of Grand Justices, the group opposing same-sex relationships drafted two referendums arguing that legal reforms should be conducted outside the Civil Code without changes in the Civil Code itself and the marriage right should be reserved only for opposite-sex couples. By contrast, the group lobbying in favor of marriage equality drafted a referendum arguing that separate legislations amount to a form of discrimination. The supporters of the legalization of same-sex relationships claimed that legalizing same-sex relationship is a medical concern. The opposing group claimed, in addition to the beliefs of the US and European opposing groups mentioned above, that the legalization of same-sex relationships would lead to a widespread outbreak of human immunodeficiency virus infection, depopulation in Taiwan, and the deterioration of traditional family values. Both the groups conducted large-scale assemblies and processions to advocate their claims and debated on mass and social media. However, lack of funds to promote their claims in public placed the supporting group at a disadvantage compared with the opposing group. The results of the vote released on 24 November 2018 were in favor of the two referendums against same-sex relationship.

Based on the results of the two referendums against same-sex marriage, the Taiwanese government enacted the Act for Implementation of Judicial Yuan Interpretation No. 748 outside the Civil Code in May 2019. This law guaranteed most but not all of the same rights entailed in a heterosexual marriage for same-sex couples [22]. For example, the Act for Implementation of Judicial Yuan Interpretation No. 748 neither legally recognize the ability of a same-sex spouse to adopt his or her partner’s non-biological children nor the ability to register transnational same-sex marriages in cases where a partner is from a country where same-sex marriage is not legalized, showing that there is still discrimination in how same-sex marriage is treated [22,23]. A study on 502 Taiwanese people conducted in December of 2019 found that 19.3% of participants have changed their views on same-sex relationship since its legalization, with most of those who have changed their views more opposed to legalization than before [24].

### 1.4. Do the Debates on the Legalization of Same-Sex Relationships Impact People’s Daily Lives?

Given that public debates on legalizing same-sex relationships are usually intense and conflicting, it is important to survey whether these debates affect people’s daily lives. Research indicates that increased exposure to same-sex relationship campaigns is associated with high stress among sexual minority individuals in the United States, and negative advertisements evoke the feeling of sadness among them [25]. Research in Australia found that more frequent exposure to negative media messages about same-sex relationship increased psychological distress in LGB individuals during the Australian Marriage Law Postal Survey [26]. Public debates may increase the chance of discussion with families, friends and colleagues and find out their opinions regarding legal recognition of same-sex relationship. The relationships with other people may be affected due to conflicting opinions on same-sex relationships. Moreover, public campaigns debating on same-sex relationships may foster a negative social climate for sexual minority individuals [27]. According to minority stress theory [28], the negative social climate may compromise quality of life of sexual minority individuals. Moreover, sleep problems and negative emotion were frequently identified among the individuals with high stress [29,30]. Therefore, negative impacts on occupational or academic performance, relationships with friends, relationships with families and mood and sleep quality resulted from public debates on legalizing same-sex relationships are needed to be examined.

Public debates on legalizing same-sex relationships lasted for 2 years before the referendums in Taiwan during the period between 2016 and 2018, hence the need for a survey on the impact of public debates on Taiwanese people, including relationships with friends and families, work, studies, and emotion. Moreover, the negative impacts on daily lives resulted from public debates on legalizing same-sex relationship may be the results of interaction between individuals and their environments. Whether gender, age, sexual orientation, personal acceptance of homosexuality, and perceived population’s acceptance of homosexuality are significantly associated with the subjective impact of public debates on daily lives warrants examination.

### 1.5. Aims and Hypotheses of the Study

The present study used data from the Investigation on the Attitude Toward Same-Sex Relationship in Taiwan. The online survey was conducted from 1 December 2018, to 31 December 2018, that is, 1 week after the referendums [31]. The present study had two aims. First, we examined the proportion of the individuals who experienced the negative impacts of public debates on the legalization of same-sex relationships on daily lives, including occupational/academic performance, family relationship, friendship, and mood/sleep quality in Taiwan. Second, we examined the associations between the negative impacts of public debates on daily lives and gender, age, sexual orientation, and personal and perceived population’s attitudes toward homosexuality. A previous study found that individuals who reported having more LGB friends scored lower on the measures of homophobia [32]. Therefore, we used the number of LGB friends as the indicator of personal attitude toward homosexuality. Given that the abundant funds to promote their claims against legalizing same-sex relationship in public placed the opposing group at an advantage compared with the supporting group, we hypothesized that non-heterosexual people experience greater negative impacts of public debates on daily lives than heterosexual people. Meanwhile, research has found that people who were females [33], younger [17] and had more LGB friends had favorable attitude toward the LGB population than their counterparts; therefore, we hypothesized that people who were females, were younger, and had more LGB friends may experience greater negative impacts on daily lives than their counterparts. Perceiving population’s favorable attitudes toward homosexuality may be positively associated with personal favorable attitudes toward homosexuality; therefore, we hypothesized that people who perceived population’s favorable attitudes toward homosexuality may experience greater negative impacts on daily lives than those who perceived population’s unfavorable attitudes toward homosexuality.

## 2. Methods

### 2.1. Participants

The method used to recruit participants is described elsewhere [31,34]. In brief, participants aged ≥20 years were recruited for the online survey through a Facebook advertisement from 1 December 2018, to 31 December 2018, that is, 1 week after the referendums. We targeted the advertisement to Facebook users based on location (Taiwan) and language (Chinese). To ensure that sexual minority individuals were recruited, we also posted the link of the Facebook advertisement to the Facebook pages of three Taiwanese health promotion and counseling centers for lesbian, gay, and bisexual individuals.

The study was approved by the Institutional Review Board of Kaohsiung Medical University Hospital (KMUHIRB-EXEMPT(II)-20160065). The study design allowed respondents to anonymously respond to the recruitment advertisement and questionnaire, and their personal information was kept confidential. The institutional review board thus waived off the requirement of written informed consent.

### 2.2. Measures

#### 2.2.1. Negative Impact of Public Debates on Legalizing Same-Sex Relationship on Daily Lives

We asked participants four questions to determine the severity of the negative impact of public debates on legalizing same-sex relationships on their daily lives in the preceding 1 week. The first question was as follows: “Currently there are many public debates regarding whether same-sex couples should be assigned legal status of unions such as marriage like opposite-sex couples. To what degree do these public debates negatively affect your occupational or academic performance/relationships with friends/relationships with families/your mood or sleep quality in daily lives?” Given that research has indicates that increased exposure to same-sex relationship campaigns is associated with high stress and negative emotion [25,26] and that sleep problems and negative emotion were frequently identified among the individuals with high stress [29,30], the present study asked the respondents about the impacts of public debates on their mood and sleep quality. Participants were asked to rate the severity of negative impact on a five-point scale: 0, none at all; 1, a little bit; 2, moderately; 3, quite a bit; and 4, extremely. For the purposes of this study, participants who scored 0 or 1 were classified into the group who experienced no or mild negative impacts; participants who scored 2 to 4 were classified into the group who experienced significant negative impacts of public debates. We used the 5-item Brief Symptom Rating Scale (BSRS-5) [35]. To test the concurrent validity of the four questions for negative impacts on daily lives. A higher total BSRS-5 score indicates having poor mental health. the results of chi-square tests indicated that respondents who experienced significant negative impacts of public debates on any domains of daily lives reported a higher total score of the BSRS-5 than those who experienced no or mild negative impacts (*p* < 0.001).

#### 2.2.2. Number of LGB Friends

We asked the following question to determine participants’ number of LGB friends: “How many LGB friends do you have?” Participants indicated their number of LGB friends on a five-point Likert scale ranging from 0 (none), 1 (very few), 2 (few), 3 (many), to 4 (very many).

#### 2.2.3. Perceived Population’s Attitudes toward Homosexuality

We asked the following question to determine participants’ perception of Taiwanese population’s attitude toward homosexuality: “To what degree does Taiwanese society accept homosexuality?” Participants indicated their perceived Taiwanese population’s acceptance of homosexuality on a five-point Likert scale ranging from 0 (very low) to 4 (very high).

#### 2.2.4. Demographic Variables

Data on participants’ gender (transgender, male and female), age, and sexual orientation were collected. For age, participants were classified into age of 20–29, 30–39, and 40 or older. For sexual orientation, participants were classified into non-heterosexual (including bisexual, homosexual, pansexual, asexual, and unsure) and heterosexual groups.

### 2.3. Statistical Analysis

Data analysis was performed using SPSS Version 20.0 (SPSS Inc., Chicago, IL, USA). Descriptive results are presented as the frequency and percentage for categorical variables and as the mean and standard deviation for continuous variables. The associations between the negative impacts of public debates on the four domains of daily lives (dependent variables) and gender, age, sexual orientation, and personal and perceived population’s attitudes toward homosexuality were examined by using multivariate logistic regression analysis. Partial regression coefficients, standard errors, Wals χ^2^, *p* value, odds ratio and 95% confidence interval were presented. Because of multiple comparisons of four domains of daily lives assessed, a *p*-value of <0.0125 (0.05/4) was considered statistically significant for all tests.

## 3. Results

### 3.1. Participants’ Negative Impact of Public Debates on Daily Lives

The data of 1370 participants were analyzed. Gender, age, sexual orientation, number of LGB friends, and perceived population’s acceptance for homosexuality with the negative impacts of public debates on legalizing same-sex relationships on daily lives are presented in Table 1. In total, 2.4% of the participants were transgender, 37.2% were male and 60.4% were female; mean age was 32.2 years (standard deviation [SD]: 9.2 years; range: 20–72); 45.9% were 20–29 years old, 34.5% were 30–39 years old, and 19.6% were 40 or older; 39.4% were heterosexual and 60.6% were non-heterosexual; 36.3% reported to have no or few LGB friends and 63.7% had many LGB friends. The mean score of number of LGB friends was 2.9 (SD: 1.3; range: 0–4). The mean score of perceived population’s acceptance for homosexuality was 1.6 (SD: 0.9; range: 0–4).

Regarding negative impact of public debates on daily lives, 39.5% of participants reported that public debates had significant impacts on occupational or academic performance; 34.2% had significant impacts on friendship; 37.7% had significant impacts on relationships with families; and 57.4% had significant impacts on mood or sleep quality.

### 3.2. Factors Related to the Negative Impacts of Public Debates on Daily Lives

Table 2 presents the results examining the factors related to the negative impacts of public debates on daily lives. The results indicated that non-heterosexual participants were more likely to report negative impacts of public debates on all domains of daily lives than heterosexual ones. Numbers of LGB friends were positively associated with negative impacts of public debates on all domains of daily lives. Males were less likely to report the negative impact on their mood/sleep quality than females. Participants who were 20–29 years old were more likely to report negative impacts of public debates on the domains of family relationship and mood/sleep quality than those who were 40 or older. Participants who were 30–39 years old were more likely to report negative impacts of public debates on the domain of mood/sleep quality than those who were 40 or older. Perceiving population’s acceptance for homosexuality were negatively associated with negative impacts of public debates on the domains of occupational/academic performance, family relationship, and mood/sleep quality.

## 4. Discussion

### 4.1. Negative Impact of Public Debates on Daily Lives

The present study showed that 57.4% of participants reported the negative impact of public debates on legalizing same-sex relationship on mood or sleep quality, and 34.2%, 37.7%, and 39.5% of participants reported negative impacts on friendship, family relationship and occupational or academic performance, respectively. Same-sex relationships have never been legalized without public debates. Intense debates on legalizing same-sex relationship between its supporters and opposers happened in public and social media and social media and significantly impacted people’s multiple domains of daily lives. Under the premise of respecting the right to express opinions, the government needs to implement the strategies to reduce the negative impacts on daily lives resulted from the public debates on law and public policies.

### 4.2. Role of Sexual Orientation for Negative Impacts

The present study showed that non-heterosexual participants were more likely to report negative impacts of public debates on all domains of daily lives than heterosexual ones. Public debates on legalizing same-sex relationships may directly interfere with non-heterosexual individuals by delivering emotionally hurtful messages. For example, a study in the United States revealed that LGB people are emotionally affected by media advertisements of marriage equality campaigns, whereas non-LGB people are unaffected; LGB people experience a 34.0% greater probability of stress with exposure to advertisements than to nonexposure to advertisements [25]. In addition, public debates on legalizing same-sex relationship may increase public attention on non-heterosexual individuals; non-heterosexual individuals had to interact with neighbors, colleagues, and family members who adopted viewpoints broadcast by the groups opposing same-sex relationships [36]. According to the social identity threat theories of stigma [37], cues from the social environment that are appraised as potentially harmful to one’s stigmatized social identity engender a threat, which in turn creates involuntary stress responses. Moreover, public debates on legalizing same-sex relationships intensified after the proposal of the referendums, and the situation was more and more disadvantageous to the legalization of same-sex relationships nearly before the vote for the referendums. Research found that LGB people reported comparatively worse life satisfaction, mental health, and overall health in constituencies with higher rates of voters saying “no” to the same-sex plebiscite [38]. The social atmosphere constructed by public debates may negatively influence non-heterosexual individuals in multiple domains of daily lives.

### 4.3. Number of LGB Friends and Perceived Population’s Acceptance of Homosexuality

The present study showed that participants who had many LGB friends were also more likely to report negative impacts of public debates on all domains of daily lives than those who had no or few LGB friends. Having more LGB friends may indicate a high level of acceptance of sexual minority. People who had favorable attitudes toward sexual minority might be vulnerable to the statements that stigmatized sexual minority people and same-sex relationship during the period of public debates. Demoralization resulted from public debates might negatively impact their multiple domains of daily lives.

Perceiving higher population’s acceptance for homosexuality were significantly associated with low risks of negative impacts of public debates on the domains of occupational/academic performance, family relationship and mood/sleep quality. Although the mechanism warrants further evaluation, it is possible that people perceiving high population’s acceptance for homosexuality may uphold the idea that debates may enhance people’s understanding of the meaning and importance of legally recognizing same-sex relationships and therefore feel less upset over the turmoil caused by public debates. However, it is also possible that people perceiving high population’s acceptance for homosexuality misunderstood the anti-LGB information embedded in the strong claims of the opposing group and thus experienced less impacts on their family relationship and mood/sleep quality.

### 4.4. Age Effect on the Negative Impacts of Public Debates

The present study found that participants who were 20–29 years old were more likely to report negative impacts of public debates on the domains of family relationship and mood/sleep quality than those who were 40 or older. Participants who were 30–39 years old were also more likely to report negative impacts of public debates on the domain of mood/sleep quality than those who were 40 or older. Research has demonstrated that young people have a more favorable attitude toward homosexuality than older people in Taiwan [17]. The favorable attitude toward homosexuality may result in shock and disappointment in response to the public debates on legalizing same-sex relationship. On the other hand, people who were 40 or older might be used to an unfavorable attitude toward homosexuality perceived from traditional societies and their friends and feel less shock during the public debates that mainly oppose legalizing same-sex relationship.

### 4.5. Gender and Negative Impacts of Public Debates

The present study found that males were less likely to report the negative impact on their mood/sleep quality than females. In Taiwan, the rate of acceptance toward the LGB population was higher in females than in males [33]. Based on the analysis of results of Taiwanese referendum, females voted on the referendum supporting legalizing same-sex relationship more than males [39]. The result of this study supported the hypothesis that females are more likely to experience negative impacts of public debates than males. There were very few transgenders responding to this study (*n* = 33); their experiences and feelings during the public debates on legalizing same-sex relationship warrant further evaluation.

### 4.6. Implication

The results of this study indicated that a high proportion of people in Taiwan suffered from negative impacts on daily lives resulted from the public debates on legalizing same-sex relationship, especially those who were non-heterosexual, friendly to sexual minority and young. Based on the results, we make recommendations to improve the spoils of public debates at both macro policy and micro individual levels. First, according to minority stress theory [22] and social identity threat theories of stigma [27], public debates on legalizing same-sex relationships may aggravate the negative social climate and interactions and thus negatively impact daily lives of non-heterosexual individuals and younger people. Whether the civil rights of sexual minority individuals can be determined through voter-initiated referendums should be comprehensively evaluated. Moreover, according to the Referendum Act in Taiwan, the government agencies should raise the position papers on the proposal of referendum to the public through public notice 90 days before the day of referendum [40]. However, the Taiwanese government refused participating the debate through national broadcast television channels and weaseled out of providing accurate information on legalizing same-sex relationship for the public. The governments should undertake the duty to protect people from being hurt by the slander disseminated by the hate groups during the debates on the issues of human right. Moreover, the governments should develop the strategies in advance to prevent negative impacts on daily lives and psychological health resulted from the public debates on the proposals of the referendum as well as from the results of the votes. The present study determined the risk groups experiencing negative impacts of the public debates and can serve as the reference of developing prevention and intervention programs for the negative impacts.

### 4.7. Limitations

This study has limitations. First, although Facebook can be used to obtain numerous participants quickly, cheaply, and with minimal effort compared with email and phone recruitment, access to Facebook is not yet universal and people are not equally motivated to use it [41]. For example, Facebook users consist of younger and more progressive people among the general population. Second, the cross-sectional study design limited the possibility of determining causal relationships between the perceived Taiwanese population’ and personal acceptance of homosexuality and the negative impact of public debates. Third, this study could not determine the mechanism by which public debates on legalizing same-sex relationship compromised daily lives. Fourth, the psychometrics of the measures used in this study warrant further examination.

## 5. Conclusions

A high proportion of people experienced the negative impacts of public debates regarding legalizing same-sex relationship on daily lives, with more profound impacts of daily lives of people who were non-heterosexual, friendly to sexual minorities, and younger. The public debates on legalizing same-sex relationships negatively impact the daily lives of various groups of people but not limited to only those of sexual minority. Prevention and intervention programs for reducing the negative impacts are necessary, especially taking the risk factors identified in this study into consideration.

## Figures and Tables

**Table 1 ijerph-17-08606-t001:** Demographic characteristics, number of LGB friends, perceived population’s acceptance for homosexuality, and negative impacts of debates on daily lives (*N* = 1370).

Variable	*n* (%)	Mean (SD)	Range
Gender			
Transgender	33 (2.4)		
Male	509 (37.2)		
Female	828 (60.4)		
Age (years)		32.2 (9.2)	20–72
20–29	629 (45.9)		
30–39	472 (34.5)		
40 or older	269 (19.6)		
Sexual orientation			
Heterosexual	540 (39.4)		
Non-heterosexual	830 (60.6)		
Number of LGB friends		2.9 (1.3)	0–4
Perceived population’s acceptance for homosexuality		1.6 (0.9)	0–4
Negative impacts of debates on daily lives			
Occupational/academic performance		1.3 (1.3)	0–4
No or mild	829 (60.5)		
Significant	541 (39.5)		
Friendship		1.1 (1.3)	0–4
No or mild	901 (65.8)		
Significant	469 (34.2)		
Family relationship		1.2 (1.3)	0–4
No or mild	854 (62.3)		
Significant	516 (37.7)		
Mood/sleep quality		1.9 (1.5)	0–4
No or mild	584 (42.6)		
Significant	786 (57.4)		

LGB: Lesbian, gay and bisexual; SD: Standard deviation.

**Table 2 ijerph-17-08606-t002:** Factors related to the negative impacts of debates on legalizing the same-sex relationship on daily lives.

Variable	Occupational/Academic Performance			Friendship			Family Relationship			Mood/Sleep Quality		
	B (SE)	Wals χ^2^ (*p*)	OR (95% CI)	B (SE)	Wals χ^2^ (*p*)	OR (95% CI)	B (SE)	Wals χ^2^ (*p*)	OR (95% CI)	B (SE)	Wals χ^2^ (*p*)	OR (95% CI)
Male ^a^	−0.481 (0.372)	1.674 (0.196)	0.618 (0.298–1.281)	−0.025 (0.371)	0.005 (0.946)	0.975 (0.472–2.016)	−0.463 (0.381)	1.477 (0.224)	0.629 (0.298–1.328)	−0.975 (0.372)	6.860 (0.009)	0.377 (0.182–0.783)
Transgender ^a^	−0.276 (0.125)	4.869 (0.027)	0.759 (0.594–0.970)	−0.036 (0.125)	0.082 (0.774)	0.965 (0.755–1.233)	−0.258 (0.127)	4.126 (0.042)	0.773 (0.602–0.991)	−0.240 (0.130)	3.406 (0.065)	0.787 (0.610–1.015)
Age of 20–29 ^b^	0.222 (0.171)	1.696 (0.193)	1.249 (0.894–1.744)	−0.255 (0.170)	2.264 (0.132)	0.775 (0.555–1.080)	0.740 (0.180)	16.917 (<0.001)	2.096 (1.473–2.982)	0.964 (0.168)	33.050 (<0.001)	2.622 (1.888–3.642)
Age of 30–39 ^b^	0.020 (0.174)	0.013 (0.908)	1.020 (0.725–1.436)	−0.245 (0.172)	2.031 (0.154)	0.782 (0.558–1.096)	0.413 (0.184)	5.017 (0.025)	1.511 (1.053–2.168)	0.498 (0.168)	8.758 (0.003)	1.645 (1.183–2.288)
Non-heterosexual ^c^	0.944 (0.136)	48.123 (<0.001)	2.569 (1.968–3.354)	0.724 (0.139)	27.182 (<0.001)	2.062 (1.571–2.707)	0.812 (0.137)	35.006 (<0.001)	2.253 (1.721–2.948)	1.054 (0.134)	62.235 (<0.001)	2.870 (2.209–3.730)
Number of LGB friends	0.174 (0.049)	12.854 (<0.001)	1.190 (1.082–1.309)	0.120 (0.049)	6.098 (0.012)	1.128 (1.025–1.241)	0.147 (0.049)	8.851 (0.003)	1.158 (1.051–1.276)	0.187 (0.048)	14.932 (<0.001)	1.205 (1.096–1.325)
Population’s acceptance for homosexuality	−0.184 (0.068)	7.394 (0.007)	0.832 (0.729–0.950)	−0.136 (0.068)	3.997 (0.046)	0.873 (0.764–0.997)	−0.310 (0.070)	19.772 (<0.001)	0.733 (0.639–0.841)	−0.299 (0.069)	18.870 (<0.001)	0.742 (0.648–0.849)

^a^: Female as the reference; ^b^: Age of 40 or older as the reference; ^c^: Heterosexual as the reference. CI: Confidence interval; LGB: Lesbian, gay and bisexual; OR: Odds ratio; SE: Standard error.

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
