# Peer review of "Impacts of Public Debates on Legalizing the Same-Sex Relationships on People’s Daily Lives and Their Related Factors in Taiwan"

_ijerph, 2020, doi:10.3390/ijerph17228606_

Round 1
Reviewer 1 Report
Introduction
Besides sleep/mood issue, explained below, the introduction is acceptable.
Methods
I would like a better explanation of how you performed multivariate logistic regression. You do not list partial regression coefficients and standard error with your results, which could prove useful. Though it appears you meet all assumptions, you could state they were checked.
How did you develop the questions? Did you pilot the questions?
MISSING! Where is the sleep/mood question in the introduction or methods? Are mood and sleep quality the same construct? Without an explanation, the chart has a serious shortcoming. I think this question should be addressed between 2.2.1-2.2.4.
Do you think asking about negative impact instead of positive impact, under 2.2.1, made a difference?
Under 2.2.2 you parsed the Likert scale differently because of skewness. Was this advisable? Did skewness make a difference, unless linearity was an issue? Did you change the meaning of the variables? When you split the scale differently, there might be a different interpretation from other variables.
Under 2.2.3, did you make the variables binary, like the other variables?
Results
Mean and SD of ages in Table 1 would be helpful. Though you made decisions binary (e.g., no/mild and significant, etc.), I would like descriptive statistics. Both would help the reader evaluate your results, and Table 1 is set to do that. I would also like results of the Likert scale at each level for each question.
Discussion
A question: Younger people who were homosexual were more aware and worried about stigma and acceptance. Is this a finding also of age, as the life trajectory changes one’s referent group from friends to self by 40? This issue could be in addition to the changing values of the younger generation.
Typo line 67 peopke should be people.
------------------------------------
Did you consider the following article?
Rich, T. S., Dahmer, A., & Brueggemann, C. (2020). Shifting perceptions of same-sex marriage in Taiwan: who changed their mind after legalization? Asian Education and Development Studies.
Author Response
Reviewer 1
Introduction
Comment 1
Besides sleep/mood issue, explained below, the introduction is acceptable.
Response
Thank you for your comment. We listed the results of previous studies (references 25-27) on the impacts of public debates on psychological stress in Introduction. We also added the description for the relationships between psychological stress and mood and sleep problems as below. Please refer to line 131-135.
“Moreover, sleep problems and negative emotion were frequently identified among the individuals with high stress [29,30]. Therefore, negative impacts on ......mood and sleep quality resulted from public debates on legalizing same-sex relationships are needed to be examined.”
Methods
Comment 2
I would like a better explanation of how you performed multivariate logistic regression. You do not list partial regression coefficients and standard error with your results, which could prove useful. Though it appears you meet all assumptions, you could state they were checked.
Response
Thank you for your suggestion. We added partial regression coefficients and standard error into the revised manuscript. Please refer to Methods section (as below, line 223-224) and Table 2.
“Partial regression coefficients, standard errors, Wals χ2, p value, odds ratio and 95% confidence interval were presented.”
Comment 3
How did you develop the questions? Did you pilot the questions?
Response
- We developed the questions based on the results of reviewing previous studies. Previous studies have found that public debates may increase the chance of discussion with families, friends and colleagues and find out their opinions regarding legal recognition of same-sex relationship (reference 27). Moreover, increased exposure to same-sex relationship campaigns is associated with high stress among sexual minority individuals (references 25 and 26). Therefore, we developed the four questions for inquiring negative impacts on occupational or academic performance, relationships with friends, relationships with families and mood and sleep quality resulted from public debates on legalizing same-sex relationships. Please refer to line 185-193. We used the 5-item Brief Symptom Rating Scale (BSRS-5) to test the concurrent validity of the four questions for negative impacts on daily lives.
- We added the result as below. Please refer to line 197-202.
“We used the 5-item Brief Symptom Rating Scale (BSRS-5) [35] to test the concurrent validity of the four questions for negative impacts on daily lives. A higher total BSRS-5 score indicates having poor mental health. the results of chi-square tests indicated that respondents who experienced significant negative impacts of public debates on any domains of daily lives reported a higher total score of the BSRS-5 than those who experienced no or mild negative impacts (p <0.001).”
3. However, the psychometrics of these four questions for negative impacts warrants comprehensive examination. We listed it as one of the limitations of this study. Please refer to line 360-361.
“Fourth, the psychometrics of the measures used in this study warrant further examination.”
Comment 4
MISSING! Where is the sleep/mood question in the introduction or methods? Are mood and sleep quality the same construct? Without an explanation, the chart has a serious shortcoming. I think this question should be addressed between 2.2.1-2.2.4.
Response
Thank you for your comment. We added introduction for the sleep/mood question in Introduction and Methods section as below.
“Moreover, sleep problems and negative emotion were frequently identified among the individuals with high stress [29,30]. Therefore, negative impacts on ......mood and sleep quality resulted from public debates on legalizing same-sex relationships are needed to be examined.” Please refer to line 131-135.
“Given that research has indicates that increased exposure to same-sex relationship campaigns is associated with high stress and negative emotion [25,26] and that sleep problems were frequently identified among the individuals with high stress and negative emotion [29,30], the present study asked the respondents about the impacts of public debates on their mood and sleep quality.” Please refer to line 189-193.
Comment 5
Do you think asking about negative impact instead of positive impact, under 2.2.1, made a difference?
Response
This comment inspired us with new ideas. Although our goal was to examine the negative impacts of public debates, there might be some people experienced positive impacts during the public debates. Further study focusing on this issue may provide various insight.
Comment 6
Under 2.2.2 you parsed the Likert scale differently because of skewness. Was this advisable? Did skewness make a difference, unless linearity was an issue? Did you change the meaning of the variables? When you split the scale differently, there might be a different interpretation from other variables.
Response
Thank you for your suggestion. We examined the level of collinearity and found that the condition index was 14.589, indicating that collinearity was not an issue. Therefore, we changed the dichotomous variable of “number of LGB friends” into the continuous variable and reanalyzed the data. The results were the same as the original ones. We revised the content of Methods, section 2.2.2. (line 206). The new results of analysis are shown in Table 2.
Comment 7
Under 2.2.3, did you make the variables binary, like the other variables?
Response
The variable of “perception of Taiwanese population’s attitude toward homosexuality” was a continuous variable.
Results
Comment 8
Mean and SD of ages in Table 1 would be helpful. Though you made decisions binary (e.g., no/mild and significant, etc.), I would like descriptive statistics. Both would help the reader evaluate your results, and Table 1 is set to do that. I would also like results of the Likert scale at each level for each question.
Response
Thank you for your suggestion. We added the means and SD of age and negative impacts on four domains of daily lives into Table 1.
Discussion
Comment 9
A question: Younger people who were homosexual were more aware and worried about stigma and acceptance. Is this a finding also of age, as the life trajectory changes one’s referent group from friends to self by 40? This issue could be in addition to the changing values of the younger generation.
Response
We agree your opinion. We added it into the revised manuscript as below. Please refer to line 318-320.
“On the other hand, people who were 40 or older might be used to unfavorable attitude toward homosexuality perceived from traditional societies and their friends and feel less shock during the public debates that mainly oppose legalizing same-sex relationship.”
Comment 10
Typo line 67 peopke should be people.
Response
Thank you for your reminding. We corrected it. Please refer to Conclusion section, line 363.
Comment 11
Did you consider the following article?
Rich, T. S., Dahmer, A., & Brueggemann, C. (2020). Shifting perceptions of same-sex marriage in Taiwan: who changed their mind after legalization? Asian Education and Development Studies.
Response
Thank you for your information. We added the result of this study as below into our revised manuscript. Please refer to line 114-117.
“A study on 502 Taiwanese people conducted in December of 2019 found that 19.3% of participants have changed their views on same-sex relationship since its legalization, with most of those who have changed their views more opposed to legalization than before [24].”
Reviewer 2 Report
This paper contributes to the literature on an important LGBTQI issue with a large N in terms of how the population of LGB are affected in Taiwan by national debates over same-sex marriage.
Minor:
1) Line 132- You can start with the outcome/DV -- "We examined the associations between (DV) and x,x,x,x
2) Line 150- I think informed consent was not waived, but rather signed informed consent? Even when anonymous participants respond, they still usually receive a copy of an informed consent.
3) Some of the fonts in the text vary in size- not sure if this was intentional, but should be removed.
4) Grammar is mostly correct, but the first paragraph on the intro should probably say "A ban" or "The ban".
Intro
5) In the intro, I'm not sure if saying "warrants" is the best choice of wording. You say it 3 times in one paragraph.
6) Under aims and hypotheses, you don't state hypotheses (or the direction).
Methods:
7) For measures, please state what the outcome variables are, and if you have multiple outcomes you are testing in different models since you say "multivariate" instead of multiple logistic regression.
Results:
8) Consider a more general title for Table 1.
Discussion:
9) The solutions/recommendations can be elaborated on much more on both macro policy and micro individual levels. Also, what the implications are for future practice and policy would be good to add.
10) Drive home the significance of this study and major findings a bit more if you can.
Author Response
Reviewer 2
Comment 1
1) Line 132- You can start with the outcome/DV -- "We examined the associations between (DV) and x,x,x,x
Response
Thank you for your comment. We revised this sentence as below. Please refer to line 150-152.
“Second, we examined the associations between the negative impacts of public debates on daily lives and gender, age, sexual orientation, and personal and perceived population’s attitudes toward homosexuality.”
Comment 2
2) Line 150- I think informed consent was not waived, but rather signed informed consent? Even when anonymous participants respond, they still usually receive a copy of an informed consent.
Response
Thank you for your reminding. We revised it into “The institutional review board thus waived off the requirement of written informed consent.” Please refer to line 180.
Comment 3
3) Some of the fonts in the text vary in size- not sure if this was intentional, but should be removed.
Response
We revised them to the same size in the revised manuscript, for example, “human rights” in line 61 and 62.
Comment 4
4) Grammar is mostly correct, but the first paragraph on the intro should probably say "A ban" or "The ban".
Response
Thank you for your comment. We revised it into “the ban.” Please refer to line 45, 47, 52, and 63
Comment 5
5) In the intro, I'm not sure if saying "warrants" is the best choice of wording. You say it 3 times in one paragraph.
Response
We changed the wording “warrants” into “…it is important to survey…” (line 120) and “…hence the need for…” (line 137).
Comment 6
6) Under aims and hypotheses, you don't state hypotheses (or the direction).
Response
Thank you for your reminding. We added the hypotheses as below into the revised manuscript. Please refer to line 155-166.
“Given that the abundant funds to promote their claims against legalizing same-sex relationship in public placed the opposing group at an advantage compared with the supporting group, we hypothesized that non-heterosexual people experience greater negative impacts of public debates on daily lives than heterosexual people. Meanwhile, research has found that people who were females [33], younger [17] and had more LGB friends had favorable attitude toward the LGB population than their counterparts; therefore, we hypothesized that people who were females, were younger, and had more LGB friends may experience greater negative impacts on daily lives than their counterparts. Perceiving population’s favorable attitudes toward homosexuality may be positively associated with personal favorable attitudes toward homosexuality; therefore, we hypothesized that people who perceived population’s favorable attitudes toward homosexuality may experience greater negative impacts on daily lives than those who perceived population’s unfavorable attitudes toward homosexuality.”
Comment 7
Methods:
7) For measures, please state what the outcome variables are, and if you have multiple outcomes you are testing in different models since you say "multivariate" instead of multiple logistic regression.
Response
Thank you for your reminding. We revised the sentence in “2.3. Statistical Analysis” and labelled the four domains of daily lives as the dependent variables as below. Please refer to line 220-223.
“The associations between the negative impacts of public debates on the four domains of daily lives (dependent variables) and gender, age, sexual orientation, and personal and perceived population’s attitudes toward homosexuality were examined by using multivariate logistic regression analysis.”
Comment 8
Results:
8) Consider a more general title for Table 1.
Response
We revised the title of Table 1 into “Demographic characteristics, number of LGB friends, perceived population's acceptance for homosexuality, and negative impacts of debates on daily lives.” Please refer to line 242-243.
Comment 9
Discussion:
9) The solutions/recommendations can be elaborated on much more on both macro policy and micro individual levels. Also, what the implications are for future practice and policy would be good to add.
Response
Thank you for your suggestion. We added a new paragraph titled “4.6. Implication” as below to elaborate our recommendations based on the results of this study. Please refer to line 330-351.
“4.6. Implication
The results of this study indicated that a high proportion of people in Taiwan suffered from negative impacts on daily lives resulted from the public debates on legalizing same-sex relationship, especially those who were non-heterosexual, friendly to sexual minority and young. Based on the results, we make recommendations to improve the spoils of public debates at both macro policy and micro individual levels. First, according to minority stress theory [22] and social identity threat theories of stigma [27], public debates on legalizing same-sex relationships may aggravate the negative social climate and interactions and thus negatively impact daily lives of non-heterosexual individuals and younger people. Whether the civil rights of sexual minority individuals can be determined through voter-initiated referendums should be comprehensively evaluated. Moreover, according to the Referendum Act in Taiwan, the government agencies should raise the position papers on the proposal of referendum to the public through public notice 90 days before the day of referendum [40]. However, the Taiwanese government refused participating the debate through national broadcast television channels and weaseled out of providing accurate information on legalizing same-sex relationship for the public. The governments should undertake the duty to protect people from being hurted by the slander disseminated by the hate groups during the debates on the issues of human right. Moreover, the governments should develop the strategies in advance to prevent negative impacts on daily lives and psychological health resulted from the public debates on the proposals of the referendum as well as from the results of the votes. The present study determined the risk groups experiencing negative impacts of the public debates and can serve as the reference of developing prevention and intervention programs for the negative impacts.”
Comment 10
10) Drive home the significance of this study and major findings a bit more if you can.
Response
Thank you for your suggestion. We revised the content of “5. Conclusion” as below to address the significance and major findings of this study. Please refer to line 363-369.
“A high proportion of people experienced the negative impacts of public debates regarding legalizing same-sex relationship on daily lives, with more profound impacts of daily lives of people who were non-heterosexual, friendly to sexual minority and younger. The public debates on legalizing same-sex relationship negatively impact the daily lives of various groups of people but not limited to only those of sexual minority. The prevention and intervention programs for reducing the negative impacts are necessary, especially taking the risk factors identified in this study into consideration.”
Reviewer 3 Report
Dear authors,
thank you for an interesting paper covering an important area connected to human (sexual) rights and public health. Overall, the paper is coherent and clear. However, I do have some questions, comments and suggestions. English is not my mother tongue, so please excuse if some of my comments are unclear.
Also, I happened to read a daily newspaper article just a couple of days ago, describing same-sex marriages in Taiwan? Perhaps an update somewhere in your paper could be of value?
Abstract
- P 1, line 23-24: The related factors – perhaps specify them here, or rephrase to related factors?
- Key words: You could add minority stress as a relevant and important key word.
The Introduction is mostly clear, but some clarifications could be made:
- P 2, line 55. The heading could be clearer. Does 2. describe Public debates on legalizing same-sex relationships, or Research on public debates and same-sex relationships?
- The section 1.3.( Public debates on legalizing same-sex relationships in Taiwan) is interesting and adds value to your study, but it completely lacks references. Would it be possible to add some support for these claims? And perhaps add the current status?
- Page 2 , line 77: In what year did the court turn down the petition? Information about that could add value for an outside reader.
- Page 3, line 118: Again: help the reader with years? Between 20XX and 20XX ??
- Page 3, line 122. Demographic characteristics is a bit vague, you could specify what you mean.
- Page 3, line 133. Clarify that previous research showed? If you add the word previous, you separate this study from your previous one, that would make it more understandable.
The Methods section is clear and explains choices made steps taken throughout. Again, some clarifications could be made:
- Page 4, line 150-151: It says that ” The institutional review board thus waived off the requirement of informed consent”. This comes across as a bit negative, from an ethical perspective. You could clarify that they waived off written informed consent and that consent was given by participants when choosing to take part in the survey?
- Page 4, line 177. Where the only options male and female? For a study with a sexual health and rights (SRHR) perspective such as this, the lack of other options is surprising. Perhaps add this as a study limitation?
The Result section is clear
The Discussion section is also mostly clear, but have some minor unclear parts:
- Why do you not elaborate on gender differences found in the result?
- Page 8, line 2-5: Language seems a bit unclear: The present study showed that over a half of participants reported the negative impact of public debates on legalizing same-sex relationship on mood or sleep quality, and over one-third of participants reported negative impacts on other three domains of daily lives.
- P 8, line 45: A word missing in the sentence?: However, it is also that people perceiving high population’s acceptance for homosexuality misunderstood the anti-LGB information
- P 9, line 54-56: The first sentence (The favorable attitude toward homosexuality may result in shock and disappointment in response to the public debates on legalizing same-sex relationship) is relevant when discussing age, the second (Admittedly, the opposing group had advantaged financial resource to promote their claims in public.) seems a bit out of place. Clarify, or delete?
- P 9. Line 67: A high proportion of peopke experienced
References appears relevant, and are well presented. Sources for part 1:3 of the Introduction could be added to strengthen the paper?
Author Response
Reviewer 3
Comment 1
Also, I happened to read a daily newspaper article just a couple of days ago, describing same-sex marriages in Taiwan? Perhaps an update somewhere in your paper could be of value?
Response
Thank you for your valuable comment. We added the information of legalizing same-sex relationship in Taiwan as below in Introduction section. Please refer to line 107-117.
“Based on the results of the two referendums against same-sex marriage, the Taiwanese government enacted the Act for Implementation of Judicial Yuan Interpretation No. 748 outside the Civil Code in May 2019. This law guaranteed most but not all of the same rights entailed in a heterosexual marriage for same-sex couples [22]. For example, the Act for Implementation of Judicial Yuan Interpretation No. 748 neither legally recognize the ability of a same-sex spouse to adopt his or her partner’s non-biological children nor the ability to register transnational same-sex marriages in cases where a partner is from a country where same-sex marriage is not legalized, showing that there is still discrimination in how same-sex marriage is treated [22,23]. A study on 502 Taiwanese people conducted in December of 2019 found that 19.3% of participants have changed their views on same-sex relationship since its legalization, with most of those who have changed their views more opposed to legalization than before [24].”
Abstract
Comment 2
P 1, line 23-24: The related factors – perhaps specify them here, or rephrase to related factors?
Response
We rephrased it into “related factors.” Please refer to line 24.
Comment 3
Key words: You could add minority stress as a relevant and important key word.
Response
We added “minority stress” as a key word of this manuscript. Please refer to line 41.
Introduction
Comment 4
P 2, line 55. The heading could be clearer. Does 2. describe Public debates on legalizing same-sex relationships, or Research on public debates and same-sex relationships?
Response
We revised it into “1.2. Advocations of Supporters and Opposers in Public Debates on Legalizing Same-Sex Relationships.” Please refer to line 58.
Comment 5
The section 1.3. (Public debates on legalizing same-sex relationships in Taiwan) is interesting and adds value to your study, but it completely lacks references. Would it be possible to add some support for these claims? And perhaps add the current status?
Response
Thank you for your suggestion. We added the references 19-24 into the revised manuscript to support these claims. Please refer to line 81-117.
Comment 6
Page 2, line 77: In what year did the court turn down the petition? Information about that could add value for an outside reader.
Response
The court turn down the petition in 1986. We added it in line 80.
Comment 7
Page 3, line 118: Again: help the reader with years? Between 20XX and 20XX ??
Response
We added “between 2016 and 2018” into the revised manuscript. Please refer to line 137.
Comment 8
Page 3, line 122. Demographic characteristics is a bit vague, you could specify what you mean.
Response
We replaced “demographic characteristics” by “gender and age” in the revise manuscript. Please refer to line 141.
Comment 9
Page 3, line 133. Clarify that previous research showed? If you add the word previous, you separate this study from your previous one, that would make it more understandable.
Response
We changed it into “a previous study” in the revise manuscript. Please refer to line 152-153.
Methods
Comment 10
Page 4, line 150-151: It says that ”The institutional review board thus waived off the requirement of informed consent”. This comes across as a bit negative, from an ethical perspective. You could clarify that they waived off written informed consent and that consent was given by participants when choosing to take part in the survey?
Response
Thank you for your comment. We revised the description into “waived off written informed consent” in the revise manuscript. Please refer to line 180.
Comment 11
Page 4, line 177. Where the only options male and female? For a study with a sexual health and rights (SRHR) perspective such as this, the lack of other options is surprising. Perhaps add this as a study limitation?
Response
There were 33 transgender people responding to our study, and their data were not included in the original analysis. We reanalyzed the data by including the 33 transgender respondents and rewrote the results. Most of the results were the same as the original ones. Please refer to Table 1 and the contents of Results section from line 229-257.
Discussion
Comment 12
Why do you not elaborate on gender differences found in the result?
Response
Thank you for your comment. We added the result of gender difference in the negative impact on mood /sleep quality as below in the revised manuscript. We also added a new paragraph discussing gender difference found in the present study as below.
“Males were less likely to report the negative impact on their mood/sleep quality than females.” Please refer to line 250-251.
“4.5. Gender and Negative Impacts of Public Debates
The present study found that males were less likely to report the negative impact on their mood/sleep quality than females. In Taiwan, the rate of acceptance toward the LGB population was higher in females than in males [33]. Based on the analysis of results of Taiwanese referendum, females voted on the referendum supporting legalizing same-sex relationship more than males [39]. The result of this study supported the hypothesis that females are more likely to experience negative impacts of public debates than males. There were very few transgenders responding to this study (n = 33); their experiences and feelings during the public debates on legalizing same-sex relationship warrant further evaluation.” Please refer to line 321-329.
Comment 13
Page 8, line 2-5: Language seems a bit unclear: The present study showed that over a half of participants reported the negative impact of public debates on legalizing same-sex relationship on mood or sleep quality, and over one-third of participants reported negative impacts on other three domains of daily lives.
Response
We rewrote these sentences as below. Please refer to line 263-266.
“The present study showed that 57.4% of participants reported the negative impact of public debates on legalizing same-sex relationship on mood or sleep quality, and 34.2%, 37.7% and 39.5% of participants reported negative impacts on friendship, family relationship and occupational or academic performance, respectively.”
Comment 14
P 8, line 45: A word missing in the sentence?: However, it is also that people perceiving high population’s acceptance for homosexuality misunderstood the anti-LGB information
Response
Thank you for your reminding. We did miss the word “possible.” We added it into the revised manuscript. Please refer to line 306.
Comment 15
P 9, line 54-56: The first sentence (The favorable attitude toward homosexuality may result in shock and disappointment in response to the public debates on legalizing same-sex relationship) is relevant when discussing age, the second (Admittedly, the opposing group had advantaged financial resource to promote their claims in public.) seems a bit out of place. Clarify, or delete?
Response
We deleted this sentence (“Admittedly, the opposing group had advantaged financial resource to promote their claims in public.”) Please refer to line 318.
Comment 16
P 9. Line 67: A high proportion of peopke experienced
Response
Thank you for your reminding. We corrected this typo. Please refer to line 363.
Comment 17
References
Sources for part 1:3 of the Introduction could be added to strengthen the paper?
Response
We added the references 19-24 into the revised manuscript to support these claims. Please refer to line 81-117.